# Evaluating the Role of the Communication Skills of Engineering Students on Employability According to the Outcome-Based Education (OBE) Theory

**Ying Wu [1], Lin Xu [1,*] and Simon P. Philbin [2]**

1   School of Foreign Languages, Northwest University, Xi'an 710069, China; 20133363@nwu.edu.cn
2   School of Engineering, London South Bank University, London SE1 0AA, UK; philbins@lsbu.ac.uk
*   Correspondence: lin_err@163.com; Tel.: +86-133-7921-7156

**Abstract:** With the advent of the digital age and Industry 4.0, the communication skills of engineering students has been exerting gradually increasing influence on the need for enhanced student employability. To address this issue, this study evaluates the influence of communication skills on the employability of engineering students from the aspects of verbal and non-verbal skills through the four key elements of Outcome-Based Education (OBE) theory. A total of 130 college engineering students and practitioners participated in the questionnaire survey, and the hypotheses were tested by regression analysis and a structural equation model (SEM). The results show that in the digital age, the verbal ($p < 0.01$) and non-verbal ($p < 0.001$) components of students' communication skills play a significant role in the employability of engineering students. Meanwhile, the employer's assessment simultaneously makes a significant contribution to the further improvement of students' communication skills, which can help educators further cultivate and improve students' communication skills through the assessment of their employability by employers. Therefore, this research has theoretical and practical significance, which not only enriches the basic theoretical knowledge, but also broadens the training methods of engineering students in the digital age, thus enhancing their employability.

**Keywords:** outcome-based education (OBE) theory; communication skills; engineering students; employability; employer's assessment

## 1. Introduction

With the advent of globalization and the digital era, society exerts greater demands upon engineering students. The employability of engineering students is no longer merely a matter of the application of technical skills, but students must also possess complementary professional skills (i.e., soft skills) to achieve personal employment and resulting career development [1]. Professional skills are the necessary accompaniment to technical skills for engineering students [2]. Such professional skills include communication skills, teamwork skills, planning and organizing skills, ICT (information and communication technology) skills, lifelong learning skills, initiative, and enterprise skills. Among these professional skills, communication skills are considered to be the most important and indispensable [3]. Nowadays, with the rapid development of digital information technology (IT) and the impact of the COVID-19 pandemic, businesses are gradually shifting towards digitalization, making remote work and online collaboration more common. In this context, communication skills have become even more important, and engineers need to be able to clearly express ideas and issues to avoid misunderstandings and errors in order to efficiently complete complex technical projects [4]. Under these circumstances, engineering students trained in the digital era need to possess strong communication skills.

After investigating the communication skills of engineering students and the opinions of employers, it has been found that many students lack the necessary workplace thinking and communication abilities, thereby making it difficult for them to tackle the challenges

of their future careers [5]. Additionally, employers have indicated that candidates lacking communication skills are at a disadvantage in the job market because this can affect work efficiency and career development; consequently, there is a requirement for universities to enhance the communication skills training of engineering students [6]. Therefore, the evaluation of students' mastery of communication skills by employers is likely to have a significant impact on their employability. In addition, the improvement of communication skills among engineering students is closely related to modern education and employer evaluation [7]. However, most existing studies have focused on the significant impact of communication skills on students' employability, and have lacked an in-depth understanding of the hierarchical levels and underlying mechanisms of how communication skills are influenced. Therefore, this study introduces three variables—namely, communication skills, employability, and employer evaluation—and focuses on the impact of engineering students' communication skills on their employability. The current changing times and digital adoption further increase the demand for students' communication skills, which also influences employer evaluation from a single technical evaluation perspective toward a comprehensive evaluation of multiple abilities. The evaluation by employers will have a reciprocal effect on the improvement of the communication skills of students and their corresponding employability. Therefore, this study aims to establish a relationship model among the three variables by utilizing a research method based on outcome-based education (OBE) theory to explore in-depth the impact of engineering students' communication skills on their employability and the underlying mechanisms.

OBE is a student-centered teaching mode, which emphasizes the judgment of students' performance through learning results, and its research parameters include knowledge, skills, and behaviors [8]. OBE theory, like other theories of learning, has been described in many ways, but the core concept is still the same—that is, what is needed to achieve and complete learning results through assessment [9]. This theory has been widely used in many countries around the world, and it also provides a framework for the education and training of engineering professionals in China [10]. Indeed, Nalla argued that—according to the OBE theory—students with certain communication skills at graduation can help improve graduate employment prospects and help reduce the gap that exists between industry and training institutions [11]. Therefore, it is feasible to adopt the OBE theory to explore the role of engineering students' communication skills on employability. From the viewpoint of objectives, the core goal of engineering education is to cultivate qualified engineering talents to meet industrial development, while the OBE theory also emphasizes from the macro perspective that talent cultivation must fit the main theme of modern education development and keep up with the situation of current economic development. From the process point of view, the OBE theory requires that the teaching design must be based on a clear outcome output and tailored to the intrinsic needs of individual students, thereby reflecting a student-centered teaching process. They theory should also build a scientific and effective learning effect assessment system to provide a scientific basis for continuous improvement and enhancement of teaching quality. This process can combine the three elements of communication skills, employability, and employer's assessment organically and form a cycle-based framework.

According to the new international context, trends of digital adoption, and the new demand for engineering professionals, this study aims to explore what kind of influence, and how, the communication skills of engineering students have on their employment skills from two aspects, i.e., verbal skills and non-verbal skills. Based on the key elements of OBE theory, the article discusses the influence and role of communication skills, employability, and employer's assessment of communication skills on employability.

Based on the concept of OBE, this study puts forward the CS-OBE model, and, according to the model, assumptions are provided, such as the direct effect of verbal skills and non-verbal skills of communication skills on employability, the direct effect of employability on employer's assessment, and the direct effect of employer's assessment on communication skills. Through the questionnaire survey method, the relevant survey

information was obtained from engineering college students and engineering practitioners in different areas of China, and the hypotheses were tested by regression analysis and structural equation modeling (SEM).

The findings of this study not only has important implications for engineering educators and engineering students, but also provides theoretical and practical implications for the development of higher engineering education. The theoretical implications are that this study combines OBE theory with the role of engineering students' communication skills and employability. Through the study, it is found that OBE theory can predict the impact of engineering students' communication skills on employability and takes the employer's assessment as a factor into account. This demonstrates the closed-loop process of the OBE theory and further identifies the applicability of the theory. The practical implications lie in the fact that this research has certain significance for engineering educators in the training of engineering students who are in line with the development of the times and who can enable students to have the required employment skills. For example, related educators can invite employers to give guest lectures at higher education institutions, which aims to let students know what skills are required to enter the workplace and enables engineering education institutions to produce workers who have skills consistent with the needs of the job market. In addition to this, engineering students can better understand the situation of employability and, according to their abilities, improve their communication skills.

## 2. Literature Review

### 2.1. OBE Theory

The concept of OBE was first introduced as part of basic educational reform in Western nations and includes competency-based education (CBE) and performance-based education (PBE). Essentially, it is competency-based education that focuses on the learner's performance and the achievement of learning outcomes [12]. The primary focus of OBE is on achieving the desired outcomes. Spady described outcomes as " . . . *what students can actually do with what they know and understand*" and which essentially involves " . . . *the visible presentation of the actual outcomes of student learning, rather than just knowing or involving purely mental processes*" [8]. Under the OBE philosophy, the teaching and learning model of courses is centered on the intended learning outcomes for students. Therefore, teaching and learning emphasizes a shift in educational philosophy and involves a student-learning orientation rather than a teaching orientation. At the same time, teaching objectives are translated into learning outcomes that students are expected to achieve, thus stimulating creativity and internal motivation for learning. Based on a review of the relevant literature, the key elements of OBE include the program educational objective, teaching design, teaching outcome, and teaching assessment [8,9].

The program educational objective (PEO) is the broad concept that defines the expected achievements of graduates within a few years of graduation and is designed according to the needs of the participants in the course [13]. Research has shown that educational objectives are a prerequisite for developing and implementing teaching designs. Only clear and reasonable objectives can guide every aspect of teaching activities, enabling the design of teaching activities to revolve tightly around the objectives. Therefore, teaching designs should support educational objectives, with scientific teaching designs linking educational objectives to corresponding teaching methods, ensuring the smooth realization of educational objectives and effectively improving teaching effectiveness. Additionally, teaching assessments can provide direct or indirect measures of students' learning. Direct assessments require students to demonstrate their achievements, while indirect assessments primarily rely on criticism. This helps assessors determine whether students have mastered or completed relevant learning tasks [14].

In one practical demonstration of the OBE theory, Rathy et al. applied the theory to the power electronics engineering curriculum and proposed a corresponding knowledge structure model [15]. Almuhaideb et al. constructed a simplified framework based on the connotations of OBE theory to foster an effective quality assurance model and effective

outcome-based education. This was based on a structure consisting of four main components, namely, strategic planning, educational practices and strategies, assessment and evaluation, and continuous improvement [16]. Li and Jiang suggested that the construction of teaching and learning models should incorporate the three core elements of OBE theory, which are defining student outcomes, designing teaching and learning activities, and developing assessment methods [13]. In 2023, a study by Wei Zheng et al. provided a sustainable teaching and learning model based on the OBE concept and the TSEM framework to enhance student engineering practice and innovation capabilities as well as cultivate applied innovative talents. [17].

In conclusion, OBE theory can effectively assess students' mastery of knowledge and skills. It can facilitate their learning, enhance the skills they need to possess, and improve their employability. Based on these, this study selects the OBE theory and discusses the role of communication skills in enhancing employability in terms of its core elements (as shown in Figure 1).

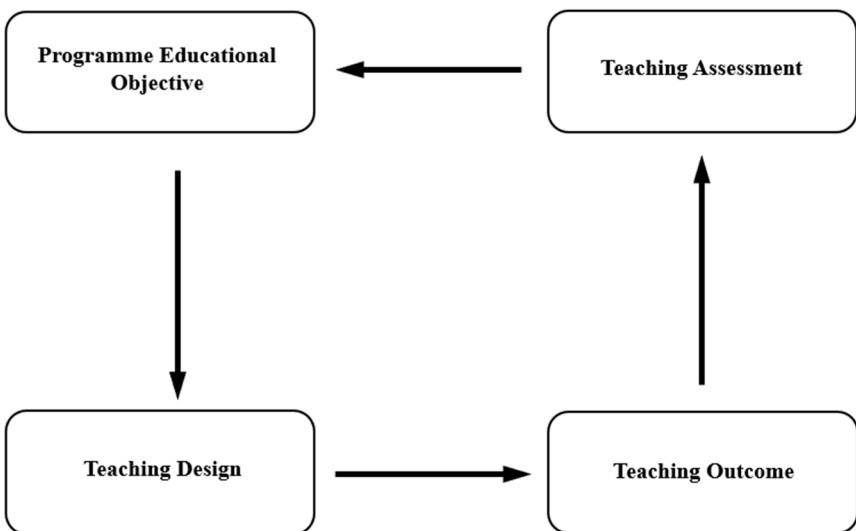

**Figure 1.** The framework of key elements of the OBE theory.

*2.2. The Connotations of Communication Skills*

Communication skills are components of employability skills and need to be developed through the learning process at universities as well as other educational levels. Communication is defined as the process by which an articulator exchanges information with a recipient in a verbal or non-verbal manner. Thus, communication skills can be divided into verbal and non-verbal skills [18,19], which are also elements of the professional skills required by students to prepare them for the workplace. In addition, communication skills can have a big impact on the success of students in their future careers [20]. In engineering projects, employee' communication skills play a critical role in project delivery [21,22]. The development of communication skills can also be considered through social constructivist learning theory [23], which focuses on learning as a social process where people construct new understandings through interaction, collaboration, and negotiation with others [24,25]. Adler et al. have shown that interaction with the environment affects students' communication strategies [26]. In addition, Yusuf and Adeoye pointed out that students can be guided to communicate and interact with other parties through verbal and non-verbal communication processes [27].

*2.3. Analysis of Engineering Students' Communication Skills Based on OBE Theory*

The OBE education concept is based on the basic requirement of "*student development as the center, competence development as the guide, and continuous quality improvement as the guarantee*". Its core elements include objectives, design, outcomes, and assessment. This concept focuses on building a training system that meets the needs of the market and

industry. Riemer discusses the importance of communication skills for novice engineering students and classifies communication skills into verbal and non-verbal dimensions [28]. Within these two dimensions, verbal skills include language communication skills, oral communication skills, and written communication skills, while non-verbal skills contain listening communication skills, visual communication skills, interdisciplinary communication skills, and intercultural communication skills (as shown in Figure 2).

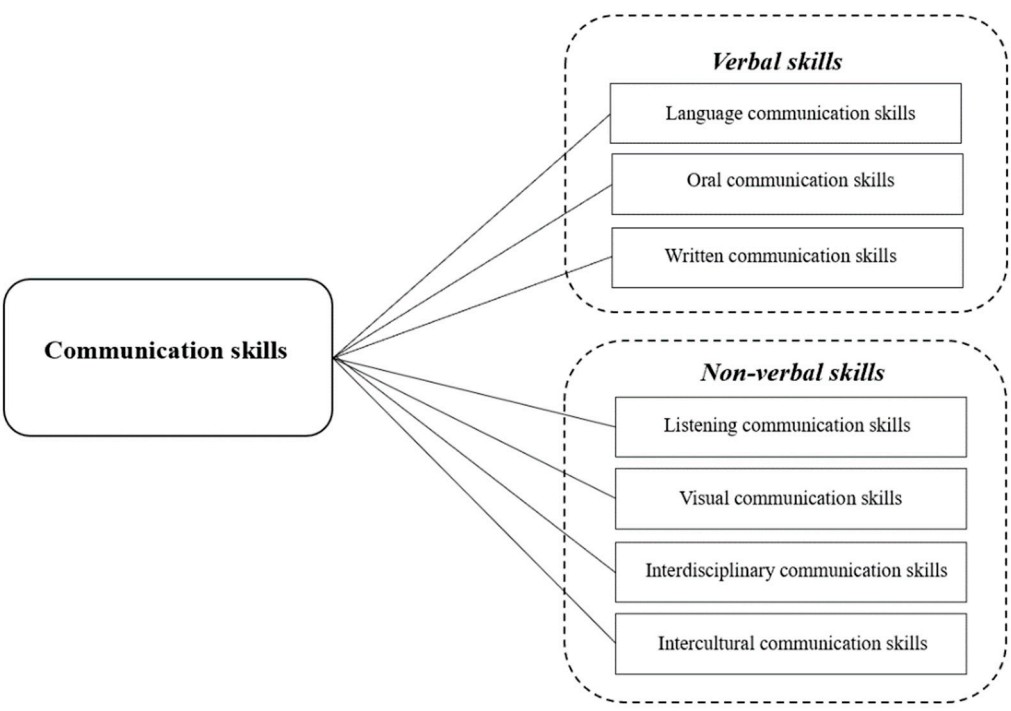

**Figure 2.** Key dimensions of communication skills.

(1)    Verbal communication skills

Strong linguistic skills and communication competences can be seen as crucial factors for advanced professional performance in a globally competitive world [29]. Linguistic competence is the skill with which a person uses ideas through spoken or written words. These skills include the acquisition of vocabulary to be able to choose the right words to convey meaning to an audience. Linguistic competence also involves the ability to organize words logically and is one of the keys to ordering knowledge [30].

- Language communication skills

In the era of globalization, international projects are on the increase, as is cross-cultural communication and cooperation, especially in international engineering practice. English is considered to be the most widely spoken language in the world [31] and can be seen as a recognized passport to improved education and employment opportunities. In addition, English has the status of a second language in almost all countries where it is not the first language, and it has become the language of international affairs and international communication as well as the language of international business and trade with its status and symbolism.

- Oral communication skills

The increasing emphasis on oral communication skills by employers has been echoed internationally across disciplines for a decade or more. Knowledge and technical know-how are important, but this knowledge must be accompanied by excellent standards of communication skills, particularly oral communication skills. Oral communication skills are considered to be one of the most important gateway skills for graduates in their new working environment [32]. Oral presentation skills are also one of the best "career

enhancers" and the biggest determinant of students' career success or failure. Methods to improve oral communication skills can include the use of class discussions, group discussions, presentations, peer reviews, role-playing, and videos of student presentations with personal feedback.

- Written communication skills

Written communication skills are a more active, rather than a passive, method of learning. Writing improves critical thinking and problem-solving skills, as well as identifying and confronting personal misunderstanding [33], and also involves the preparation of materials for different applications. Graduated engineering students often report that the workload of written communication has increased over time. Indeed, a study in Poland found that engineering students continued to believe that written communication is more difficult than oral communication, despite having completed various written tasks (e.g., lab reports, projects, etc.) before graduation [34].

(2) Non-verbal communication skills

Non-verbal communication is the process of sending and receiving information without the use of spoken or written words. It is also known as a standard language. Many people do not only employ words to communicate with others but can also include non-verbal elements such as physicality and visions [35].

- Listening communication skills

Kline further affirms the importance of listening in the workplace [36], emphasizing that listening skills are crucial in all industries including engineering. Listening is the ability to accurately receive and interpret information during the communication process, and, without effective listening skills, verbal messages may be distorted or ignored, leading to a failed communication process. Listening skill practice can be integrated into the learning environment to help students carry out team-based assignments, an increasingly important task in engineering.

- Visual communication skills

Visual communication skills refer to the use of visual aids such as drawings, illustrations, photographs, symbolic pictures, diagrams, pictograms, and other such items to share engineers' ideas, thoughts, or insights. It is clear that every area of the engineering profession relies heavily on the use of visual forms as a means of non-verbal communication. Visual literacy requires the ability to perceive visual images, including icons, and to make sense of them, and it also means being able to produce images that can be used in communication. This is as crucial to the future of design engineers as the ability to listen and speak in verbal communication [37]. Understanding visual images are often quicker than understanding their written counterparts, and the hieroglyphic nature of visual images facilitates communication between people with different linguistic and cultural backgrounds. As an engineer in the new age, one needs to consider how to read, write, and learn images, as well as how to design new ones [38,39].

- Interdisciplinary communication skills

Communication and learning between disciplines are also important aspects of communication skills that require enhancement. Interdisciplinary communication skills are increasingly recognized as essential skills for engineering graduates, especially those who intend to work internationally. Not only do engineers need to communicate and work with fellow engineers from other countries, but they are also required to master effective communication across disciplines, such as engineering and management. Interdisciplinary projects in university education can contribute to the development of students' teamwork skills. The integration of these programs has been partly due to the industry's need for engineers with more extensive skills. With interdisciplinary communication skills, engineers can build strong working relationships between team members, and a positive working environment can be created [40].

- Intercultural communication skills

The emergence of new ICT (information and communications technologies, e.g., internet, email, satellite communications), faster and cheaper international transport, migration flows, and the internationalization of business have led to an increasing number of people—including engineers—engaging in cross-cultural exchanges. Moreover, the internationalization of degree courses is not only important for remaining competitive in a global economy but are also indispensable in a world where survival can only be achieved through global cooperation. A prerequisite for successful global cooperation is the development of intercultural competence. In an era of globalization, engineers need to communicate with people from home and abroad as well as from different cultural backgrounds. Research shows that intercultural communication skills enable students to adapt to different cultures and deal with different social systems.

The key dimensions of communication skills are shown in Figure 2.

### 2.4. Research Gap

To sum up, at present, researchers have conducted studies on the relationship between OBE theory and communication skills and its influencing factors, but there remain certain research deficiencies in the knowledge base:

(1)  Previous research has focused on the application of OBE theory in developing engineering students' basic communication skills, such as in English and writing courses, rather than researching the application of communication skills in professional contexts and practices. This has left the situation unclear as to how OBE theory can enhance advanced communication skills and practical abilities.

(2)  While some studies have explored the impact of communication skills on employability, few have systematically assessed how different levels of communication skills affect employability. Research has also rarely employed multi-level research methods for examination. Consequently, this limits our precise understanding of the roles and mechanisms of influence at different levels of communication skills.

(3)  Existing research has mainly discussed this issue from the student perspective, with little consideration of the employer's perspective and needs. Moreover, research has focused on exploring a single path of influence and has failed to comprehensively explain the combined effects of multiple influencing variables. This has also led to incomplete research conclusions.

## 3. Hypotheses and Framework

### 3.1. The Relationship between Verbal Skills and the Employability of Engineering Students

One of the goals of higher educational institutions is to prepare graduates for the workplace upon graduation. Despite this, many traditional programs continue to teach traditional courses at the cost of ignoring the knowledge and skills needed for today's job market and that of the future. The literature confirms that verbal skills of communication are considered important by both employers and students in the workplace [41].

Verbal skills of communication have become indispensable skills in the workplace. Traditionally, communication refers to spoken communication between people, but the digital age has changed this monolithic form of communication, which also includes language, oral, and written communication skills. The digital age has pushed back the limits of time and space, providing unlimited convenience for students to learn with new media tools. Communication tools such as email and web-based chat forums allow for the real-time sharing of documents and information and maintaining constant contact, and enable real-time conversations aimed at solving problems and making decisions. This approach increases students' motivation to learn and thus facilitates the achievement of learning outcomes [42,43]. Today's workplaces are diffuse places where face-to-face communication in a room is not always feasible. The transition to remote working for engineering teams has become even more necessary, especially with the impact of the COVID-19 pandemic. Web conferencing and virtual meeting tools provide chat, audio,

and visual communication capabilities for engineers and other workers, thereby enabling teams to work together virtually. The use of these new media also puts forward new communication skill requirements for engineering graduates in the Industry 4.0 era.

For example, Santos et al. believe that language skills are related to the development of communication skills by learning how to understand, speak, and write correctly, and made an in-depth assessment of the influence of communication skills on employability through English language courses for three consecutive years [44]. The results show that oral and written communication skills are the final soft skills in verbal communication skills, and the research of the investigators shows that students think oral presentation is more difficult than completing written tasks.

Digital communication platforms promote students' communication, and it is the main medium for engineering students to communicate verbally. In addition to the use of new media, digital education also can be seen as an emerging strategy to promote reform of the education system to improve the educational capacity and, therefore, employability of engineering students [45]. However, existing research has found that more than half of the world's journals in advanced literature are published in English as the common language. As a result, it has become necessary for engineering students to develop their English language skills, which can lead to more career opportunities, such as learning about cutting-edge knowledge, understanding developments in the discipline, accessing opportunities to study abroad, ultimately enhancing their employability. What is more, digital developments are also placing new demands on the language communication skills of engineering students, and in today's fast-paced world, it has become particularly important to work concisely and efficiently. The format of engineering communications varies considerably, from order letters to letters of inquiry, from technical proposals to calculation summaries, from official technical memorandum to analysis reports, and from product specifications (manuals) to slide presentations. All of these require that technical communications are not only accurate, clear, and appropriate, but that their language is also clear and easy to understand [46].

In summary, the verbal skills of engineering students have a profound impact on their employability. When students have good English language skills, oral communication skills, and writing skills, they can perform well in relevant projects and increase their motivation; conversely, if engineering students have poor English language skills, they are unable to communicate well in the workplace and are unable to complete relevant tasks and solve problems through written or textual communication, thus decreasing their motivation. In contrast, students will be less motivated if they are unable to communicate well in English in the workplace and complete relevant tasks and solve problems through written communication.

Therefore, this study proposes the following hypotheses:

**H1.** *Verbal skills have a positive effect on the employability of engineering students.*

**H1a.** *Language communication skills have a positive impact on the employability of engineering students.*

**H1b.** *Oral communication skills have a positive impact on the employability of engineering students.*

**H1c.** *Written communication skills have a positive impact on the employability of engineering students.*

*3.2. The Relationship between Non-Verbal Skills and the Employability of Engineering Students*

Non-verbal communication is a process that uses actions other than words to produce meaning and includes all communications other than speech [47]. The non-verbal cues used by any speaker can make a greater impression on others than material that is merely delivered verbally. In this sense, the same non-verbal communication has a plural meaning in multiple cultures. Studies have shown that non-verbal skills also have a significant role in the workplace [48]. In the absence of close coordination between verbal and non-verbal communication, non-verbal skills are the dominant factor to be considered. For example, by employing images, even without verbal descriptions, non-verbal content

will indicate real meaning through behaviors in a given situation [49]. In addition to linguistic signs and symbols, for engineering students, the ability to communicate across disciplines and across cultures to convey information is also an indispensable element of non-verbal communication.

According to the classification of non-verbal skills, they mainly include listening, visual, interdisciplinary, and intercultural communication skills. Among these, listening is considered to be a factor in non-verbal skills [50]. In a listening session, students are not passive recipients who remain silent and submissive during the exchange of information, but active recipients who express their thoughts and feelings verbally and non-verbally [51]. Employability is enhanced when students actively listen and engage in a range of learning activities, including personal development plans, so listening is key to learning and improving employment opportunities for post-college undergraduates. In this listening process, employability is most effectively taught through active, engaged learning, providing opportunities for work experience, and encouraging reflection [52].

The visual communication skills of non-verbal skills are used to motivate engineering students to enhance their communication skills through the learning of digital technology. Howard and Mozejko classify the three "eras" of digital technology convergence as pre-digital, the personal computer, and the internet [53], which have exacerbated changes in the labor market. Today, the engineering sector faces many pressing challenges, including increasing productivity levels, coping with high levels of fragmentation and complexity, and taking advantage of emerging opportunities in digital transformation [54]. For engineering students, one of the biggest differences between the study of engineering and that of other professions is the need for engineering professionals to learn engineering drawing. While traditional engineering drawing was limited to drawing on plans, the digital age has seen a profound change in this. Digitalization has rendered certain job tasks obsolete but has also brought employment opportunities, with new, flexible forms of work increasing the requirements to become (or remain) employable in the digital economy and fostering a strong culture of lifelong learning [55]. For example, students can use CAD (computer-aided design) to establish a link between the two-dimensional "drawing" and the three-dimensional "form" [51]. A similar case is that of the chemical industry, where a truly digital product is the AUROOMTM certification platform [56]. This digital platform allows designers to access an immersive digital platform and a realistic database of virtual car colors.

Non-verbal communication skills also include interdisciplinary communication skills and, in the era of Industry 4.0, embedding elements of interdisciplinary collaboration into team learning is also conducive to strengthening students' collaboration and communication skills, thus promoting their employability [57]. To develop a fully stimulating approach to enhancing students' employability, it makes sense to provide training in interdisciplinarity in universities [58]. For example, BIM (Building Information Modeling) as a new platform for assistance and communication links design phases that are initially separated due to different professions to enhance productivity [59]. For manufacturing, the integrated status of skills related to smart manufacturing, information technology, data analysis, and systems integration will create greater demands as the process of digitalization continues to deepen. Due to the collaborative nature of smart manufacturing production technologies, the smooth implementation of products and processes requires complex and efficient coordination between various mechanical, hydraulic, electronic and software components. As a result, the industry needs more "versatile architects and versatile engineers" with comprehensive management skills such as broad and complex knowledge structures, as well as holistic competencies, creativity and the ability to learn sustainably [60].

Besides interdisciplinary communication skills, intercultural communication skills are also important for engineering students since multinational business activities have developed rapidly around the world and more and more multinational joint ventures have undertaken construction projects and operating across borders [61]. This background, however, not only brings benefits to the enterprises, but also spawns various risks and

challenges, which may cause significant economic losses or even political events if they are careless, and seriously affect the multinational business activities of the enterprises. Other employees who are not familiar with the cultural background will habitually understand this information according to their cultural background when receiving them, which will lead to a potentially divergent understanding of the same piece of information by both sides; thus, misunderstandings and even conflicts will ensue. If such misunderstandings are not eliminated in time, they can have many negative effects and the people concerned may lose their jobs and have their employment affected as a result. In addition to this, cross-national and cultural competencies also enable engineering talents to work more effectively in international teams and leadership/management [62].

In summary, listening communication skills, visual communication skills, interdisciplinary communication skills, and intercultural communication skills of non-verbal skills have an influential role in the employability of engineering students. Under the intersection of the digital era, engineering students who can obtain active listening skills, have certain professional skills, and understand different backgrounds and cultures can bridge the ability gap brought by the digital era, improve their work efficiency and obtain more opportunities in the workplace.

Therefore, this study proposes the hypotheses:

**H2.** *Non-verbal skills have a positive impact on the employability of engineering students.*

**H2a.** *Listening communication skills have a positive effect on the employability of engineering students.*

**H2b.** *Visual communication skills have a positive impact on the employability of engineering students.*

**H2c.** *Interdisciplinary communication skills have a positive impact on the employability of engineering students.*

**H2d.** *Intercultural communication skills have a positive impact on the employability of engineering students.*

*3.3. The Relationship between the Employability of Engineering Students and Employer's Assessment*

Global digitalization has not only changed the world's economic horizon, but also its social horizon. Digital technologies are currently transforming the relationships between economic players in areas such as energy, construction, banking, transport, retail, education, healthcare, media, and security. The number of jobs requiring ICT (information and communication technology) and digital literacy is increasing rapidly and the ability to use these technologies is becoming a major staffing requirement [63].

In this context, the employability of engineering students is also different from how it used to be. More than 20 years ago, Hillage and Pollard proposed a model of employment based on four components for obtaining and continuing to maintain a job, including assets (i.e., knowledge, skills, and attitudes), deployment (i.e., career management and job search skills), presentation (i.e., resume, cover letter, interview), and personal/external environment (i.e., responsibilities, family situation, labor market supply, and demand) [64]. In 2003, Knight and Yorke created the USEM frameworks (understanding, skilled practice, efficacy beliefs, and meta-cognition) to embed employability into the curriculum [65]. These frameworks seek to produce work-ready graduates by bridging the gap between educational institutions and the workplace. The assessment of engineering professionals is becoming more diversified so that they are evaluated on a combination of competencies rather than just technical knowledge. This covers problem-solving, decision-making, organizational and time management, communication ability, interpersonal skills, leadership and influence, creativity, innovation, flexibility and ability to conceptualize, lifelong learning, professional behavior, motivation, and personal strength [66].

Briggeman and Norwood found in a web-based survey of more than 400 college graduate employers that, due to changes in science and technology and related training institutions, the assessment of engineering students' employability has become increasingly

diverse, with employers using a variety of signals to determine the employability attributes of candidates, including coping skills, personality, communication skills, problem-solving skills, and the ability to work with others, with communication skills considered to be the most important competency [67].

In summary, the evolving development of the times has placed new demands on the employability of engineering students, and such demands have further altered the employer's assessment of engineering students, with market demand shifting from previously technical talents to versatile talents in line with the digital age. Therefore, this paper proposes the hypothesis that:

**H3.** *Engineering students' employability has a positive impact on employers' assessment.*

*3.4. The Relationship between Employer's Assessment and Engineering Students' Communication Skills and Employability*

Today, universities are not only focusing on the academic achievements of students in preparing engineering students to meet market demands, but also on the general skills or "soft skills" that students need to compete in the global marketplace. More importantly, employers are now placing a high value on general skills and personality when selecting their future employees. Maxwell explored the perceptions of employers on the employability skills of human resource management students. Communication and problem-solving emerged as the core skills, followed by decision-making, teamwork, independent working, leadership, personal learning and development, strategic skills, information retrieval, and numerical skills [68,69]. Based on the employer's assessment of the employability of engineering students, schools are making several changes to their education and training methods to improve the communication skills of engineering students and to produce students who meet the needs of the market and industry. For example, universities are transforming their automotive electrical and electronic courses based on the digital age model, not only by developing students' basic knowledge of automotive electrical and electronic theory, but also by organizing the courses with a focus on theoretical foundations and training processes through practical tasks, workflows, and work situations [70].

Haskova et al. (2021) argue that an employer's assessment can directly assess the quality of higher vocational education and training, where employers believe that the available personnel do not meet their requirements for the knowledge and skills of potential employees. In order to ensure a sustainable reflection of employers' requirements and needs for secondary vocational education and training, an online-communication platform was provided that organically integrates school training with employers' assessment [71].

In addition, based on employers' assessment of engineering students' employability, relevant schools are making a series of changes in education and training methods to improve engineering students' communication skills and to train students who meet the needs of the market and industry. For example, universities are transforming their automotive electrical and electronics curricula based on the digital age model, not only by developing students' basic knowledge of automotive electrical and electronics theory but also by organizing the curriculum with a focus on theoretical foundations and training processes through practical tasks, workflows, and work situations. This way allows students to develop knowledge and professional skills in their professional actions, developing their professional, methodological, and communication skills and equipping them with the full range of professional skills needed for their future work. By sharing cutting-edge knowledge related to teaching and learning with students promptly, students can learn independently and on the move, increasing after-class interaction with students and motivating them to learn. Classroom teaching makes full use of modern teaching tools such as objects, videos, and slides to improve the effectiveness of classroom teaching and enhance students' perceptual understanding and communication skills.

In summary, the employer's assessment has a significant effect on the teaching and learning development model in higher education, and schools can adjust their teaching methods according to the employer's assessment of students' communication skills in

order to improve students' employability. Therefore, the following hypothesis is proposed in this paper:

**H4.** *Employer's assessment positively moderates the role of engineering students' communication skills on employability.*

Based on the above analysis, this study constructs a conceptual model of communication skills supported by the adoption of the OBE theory, as shown in Figure 3.

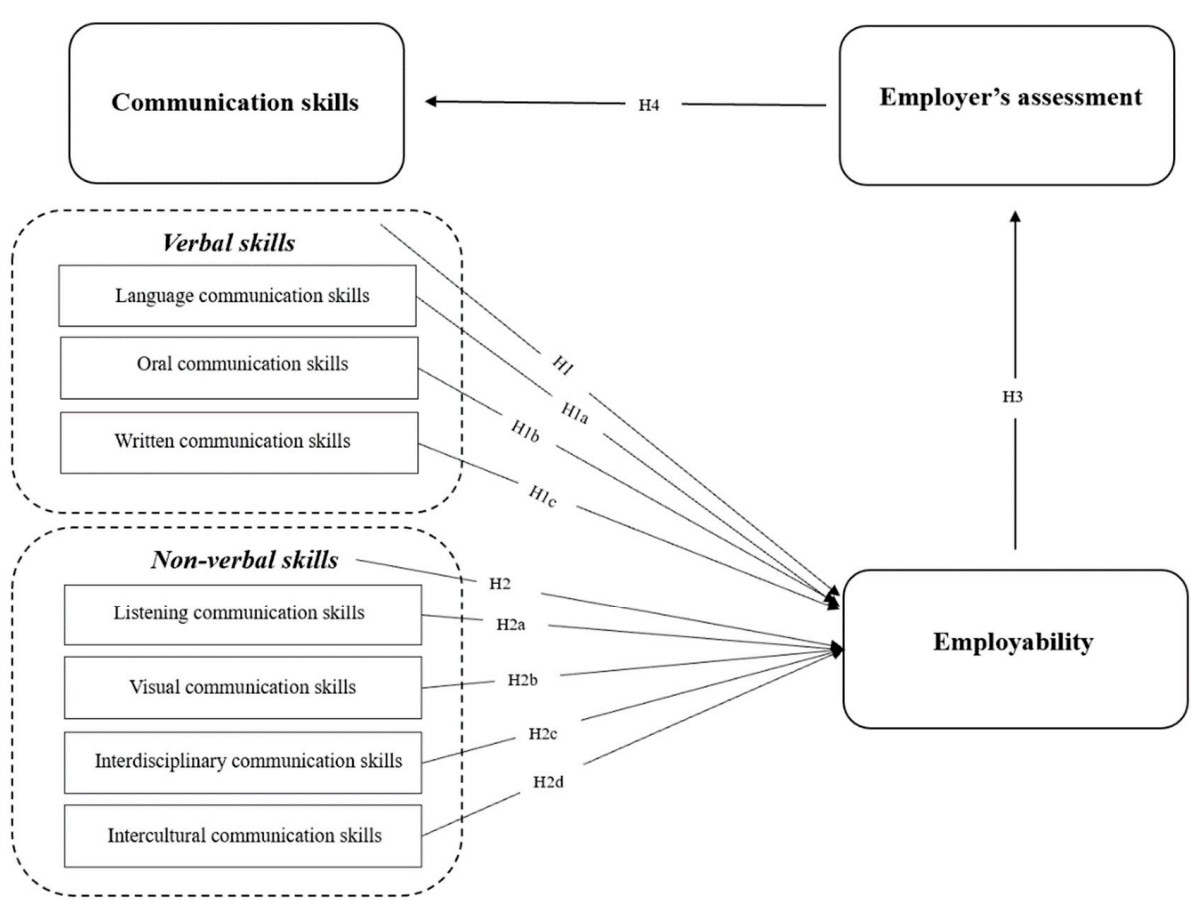

**Figure 3.** OBE theory model for communication skills.

## 4. Method

### 4.1. Design and Generation of Sample Tables

The questionnaire method has the characteristics of lower data-collection costs, stronger feasibility, and higher external validity when compared to many other methodologies. A well-formed questionnaire can be brief and takes minimal time to fill out. Simultaneously, researchers collect a great deal of data within a short time through a questionnaire survey, thus reducing the cost of data collection. In terms of operability, researchers can choose appropriate measurement tools, and design data collection schemes and sampling procedures according to the research contents and core variables, to distribute and collect questionnaire data through network channels. In terms of external validity, the subjects answers questions according to their real feelings and experiences in actual work and life, and the data has strong applicability in work and life scenarios, which improves the external validity of the research.

This paper mainly investigates the relationship between communication skills and the employability of engineering students. In order to further understand the relationship among students' communication skills, employability, and employer's assessment, the variables involved in this study are first selected from mature measurement scales, and the measurement items of each variable are specially designed in accordance with the principle

of accuracy and conciseness. Furthermore, the validity of measurement items and the reliability of the instrument is underpinned through selecting the measurement variables from the extant literature. Based on the selected items of the scale, faculty members in engineering at universities were asked to adjust the measurement items of variables for the purpose of ensuring the validity and reliability of the measurement items. By systematically sorting out the revised opinions, the measurement items of the research variables were supplemented and improved, and the language expression of measurement items was refined so that the respondents can understand them more easily.

This questionnaire consists of two parts: the first part investigates the demographic information, including the gender and grade of the respondents; the second part investigates the role of communication skills on the employability of engineering students based on the OBE theory, with a total of 25 questions. All questions were scored from 1 to 5 on a 5-point Likert scale, where 1 means "strongly disagree" and 5 means "strongly agree". Appendix A provides the structure of the questionnaire and the sources of the questions.

*4.2. Measurement*

The target population of this study was engineering university students and engineering practitioners in different regions of China. The survey was designed and conducted from 8 to 12 August 2022 by using Sojump to distribute questionnaires to engineering students. To ensure the authenticity and objectivity of the survey data, all participants were informed that their answers would be anonymous and the purpose and importance of the survey, as well as how the results of the questionnaires were only to be used for academic research, guaranteeing the privacy and security of the participants. This was performed to receive as many valid responses as possible. After the processes of matching and sorting, invalid questionnaires were eliminated, including invalid questionnaires with more consistent answers to question items, or invalid questionnaires with contradictory answers before and after, and finally 130 valid matching questionnaires were obtained, with a valid response rate of 65%. The basic demographic characteristics of the participants in this study are shown in Table 1.

**Table 1.** Profile of respondents.

| Demographic Variable | All (N = 130) | |
|---|---|---|
| | Frequency (%) | |
| *Gender* | | |
| Male | 47 | 36.15 |
| Female | 83 | 63.85 |
| *Age* | | |
| 18–21 | 13 | 10 |
| 22–25 | 79 | 60.7 |
| 26–35 | 38 | 29.3 |
| *Education* | | |
| Bachelor's | 32 | 24.62 |
| Master's | 53 | 40.77 |
| Doctor | 10 | 7.69 |
| In work | 35 | 26.92 |
| *Region (Province of China)* | | |
| Shaanxi | 61 | 47% |
| Henan | 13 | 10% |
| Shanxi | 4 | 3% |
| Hebei | 6 | 4.6% |
| Gansu | 1 | 0.8% |
| Chongqing | 3 | 2.3% |
| Sichuan | 10 | 7.7% |
| Hubei | 1 | 0.8% |

**Table 1.** *Cont.*

| Demographic Variable | All (N = 130) | |
|---|---|---|
| | Frequency (%) | |
| Guizhou | 1 | 0.8% |
| Jiangxi | 1 | 0.8% |
| Hunan | 1 | 0.8% |
| Tianjin | 1 | 0.8% |
| Shandong | 1 | 0.8% |
| Qinghai | 1 | 0.8% |
| Jiangsu | 3 | 2.3% |
| Zhejiang | 2 | 1.5% |
| Guangdong | 3 | 2.3% |
| Shanghai | 11 | 8.5% |
| Beijing | 2 | 1.5% |
| Xinjiang Uygur Autonomous Region | 1 | 0.8% |
| Inner Mongolia Autonomous Region | 2 | 1.5% |

In regard to the demographic data, 36.15% of the participants were male and 63.85% were female. A total of 7.69% were doctoral students, 40.77% were master's students, 24.62% with a bachelor's degree, and 26.92% were already in the workforce. The figures indicate that the overall education level of the respondents to this survey is high. The mean age of these respondents was 25 years old [standard deviation (SD) = 2.4] and the range was 18–35 years old, with 10% being 18–21 years old, 60.7% being 22–25 years old, and 29.3% being 26–35 years old. The study participants were representative of the target population and were mainly engineering students and workers from universities in different regions of China, with a wide regional spread.

*4.3. Data Analysis*

Before starting the data analysis, all completed questionnaires (N = 130) were checked for completeness and more than 5% of missing items, as recommended by Seo, in order to ensure the quality of the data. All questionnaires were answered completely. Each item was coded as favorable or unfavorable. Items with a score below 3 represent an unfavorable response; otherwise, it means favorable [72].

**5. Results**

*5.1. Descriptive Statistics and Correlation Analysis*

The correlations between all dimensions can be viewed in Appendix B, which highlights that all variables are significantly correlated ($p < 0.05$) and none of the correlation values exceed the threshold of 0.9. This suggests that there is no multicollinearity between items [73].

*5.2. Testing the Measurement Model—CFA*

The reliability of the variables can be tested by application of Cronbach's Alpha coefficient. If alpha > 0.6, it demonstrates that the scale is valid, and the larger the alpha reliability coefficient indicates better reliability of the scale. SPSS software was used and the results of the reliability test are shown in Appendix C. This shows that the alpha values for all variables are greater than 0.7 and the Corrected Item-Total Correlation (CITC) values for all measures are greater than 0.3, thereby indicating that the data of each variable meet the requirements of reliability.

(1) Verbal skills involve three skills. Language communication skills have four items and Cronbach's alpha is 0.930. Oral communication skills have four items and Cronbach's alpha is 0.833. Written communication skills have three items and Cronbach's alpha is 0.808.

(2) Non-verbal skills involve four skills. Listening communication skills have two items and Cronbach's alpha is 0.799. Visual communication skills have two items and Cronbach's alpha is 0.861. Interdisciplinary communication skills have three items and Cronbach's alpha is 0.881. Intercultural communication skills have two items and Cronbach's alpha is 0.931.

(3) Employability has three items and Cronbach's alpha is 0.863.

(4) Employer's assessment has two items and Cronbach's alpha is 0.911.

Then, exploratory factor analysis (EFA) was adopted to test the validity of the factors. The results of EFA are presented in Table 2. From Appendix D, it can be observed that the Kaiser–Meyer–Olkin (KMO) values of the study variables are greater than 0.7 and Bartlett's spherical test significance is less than 0.001, which is statistically significant. Meanwhile, Table 2 shows that the factor loading of each question item is greater than 0.5, the Construct Reliability (C.R.) value is greater than 0.7, and the Average Variance Extracted (AVE) value is greater than 0.5. It can be observed that each question item belongs to the corresponding variable or dimension and has good discriminant validity and convergent validity that can be tested by SEM.

**Table 2.** The results of factor analysis.

| Construct | Items | Factor Loading | C.R. | AVE |
|---|---|---|---|---|
| Language communication skills | LCS1 | 0.832 | 0.937 | 0.789 |
| | LCS2 | 0.895 | | |
| | LCS3 | 0.929 | | |
| | LCS4 | 0.895 | | |
| Oral communication skills | OCS1 | 0.794 | 0.831 | 0.559 |
| | OCS2 | 0.840 | | |
| | OCS3 | 0.794 | | |
| | OCS4 | 0.520 | | |
| Written communication skills | WCS1 | 0.808 | 0.818 | 0.600 |
| | WCS2 | 0.736 | | |
| | WCS3 | 0.778 | | |
| Listening communication skills | LSCS1 | 0.876 | 0.793 | 0.659 |
| | LSCS2 | 0.743 | | |
| Visual communication skills | VCS1 | 0.874 | 0.858 | 0.752 |
| | VCS2 | 0.861 | | |
| Interdisciplinary communication skills | IDCS1 | 0.892 | 0.891 | 0.732 |
| | IDCS2 | 0.793 | | |
| | IDCS3 | 0.878 | | |
| Intercultural communication skills | ICCS1 | 0.942 | 0.945 | 0.896 |
| | ICCS2 | 0.951 | | |
| Employability | EM1 | 0.841 | 0.877 | 0.704 |
| | EM2 | 0.853 | | |
| | EM3 | 0.823 | | |
| Employer's assessment | EA1 | 0.888 | 0.890 | 0.802 |
| | EA2 | 0.903 | | |

Notes: C.R. = Construct Reliability. AVE = Average Variance Extracted.

### 5.3. Regression Results

$R^2$ is a measure of the goodness-of-fit of a linear model and tends to be an overestimate of population parameters which range from 0 to 1 [74]. The closer to 1, the better the goodness of fit of a linear model. The standard error of the estimate is the square root of the residual mean square and measures the spread of the residuals about the fitted line, so it is also a measure of goodness of fit of a linear model [74]. The closer to 0, the better the goodness of fit of a linear model. The results (as shown in Table 3) demonstrate that $R^2$ in the three models are 0.665, 0.671, and 0.585 and the adjusted $R^2$ are 0.659, 0.652, and 0.575 in the three models, respectively. The F values are equal to 125.77, 35.60, and 59.27, which are also highly significant ($p < 0.001$).

**Table 3.** Regression analysis.

| | Employability | | Employer's Assessment |
|---|---|---|---|
| | Model 1 | Model 2 | Model 3 |
| Intercept | 0.695 ** (0.234) | 0.627 * (0.247) | 0.269 (0.316) |
| Verbal skills | 0.262 ** (0.098) | | 0.139 (0.131) |
| Non-verbal skills | 0.588 *** (0.101) | | 0.427 ** (0.148) |
| Language communication skills | | 0.057 (0.060) | |
| Oral communication skills | | 0.200 *(0.087) | |
| Written communication skills | | 0.012 (0.101) | |
| Listening communication skills | | 0.114 (0.091) | |
| Visual communication skills | | 0.194 * (0.079) | |
| Interdisciplinary communication skills | | 0.125 (0.100) | |
| Intercultural communication skills | | 0.167 * (0.080) | |
| Employability | | | 0.391 *** (0.115) |
| $R^2$ | 0.665 | 0.671 | 0.585 |
| Adj. $R^2$ | 0.659 | 0.652 | 0.575 |
| F | 125.77 *** | 35.60 *** | 59.27 *** |

Note: Unstandardized regression coefficients are displayed, with standard errors in parentheses. * $p < 0.05$. ** $p < 0.01$. *** $p < 0.001$.

*5.4. SEM*

The hypotheses were examined and the results of path analysis were represented in Figure 4.

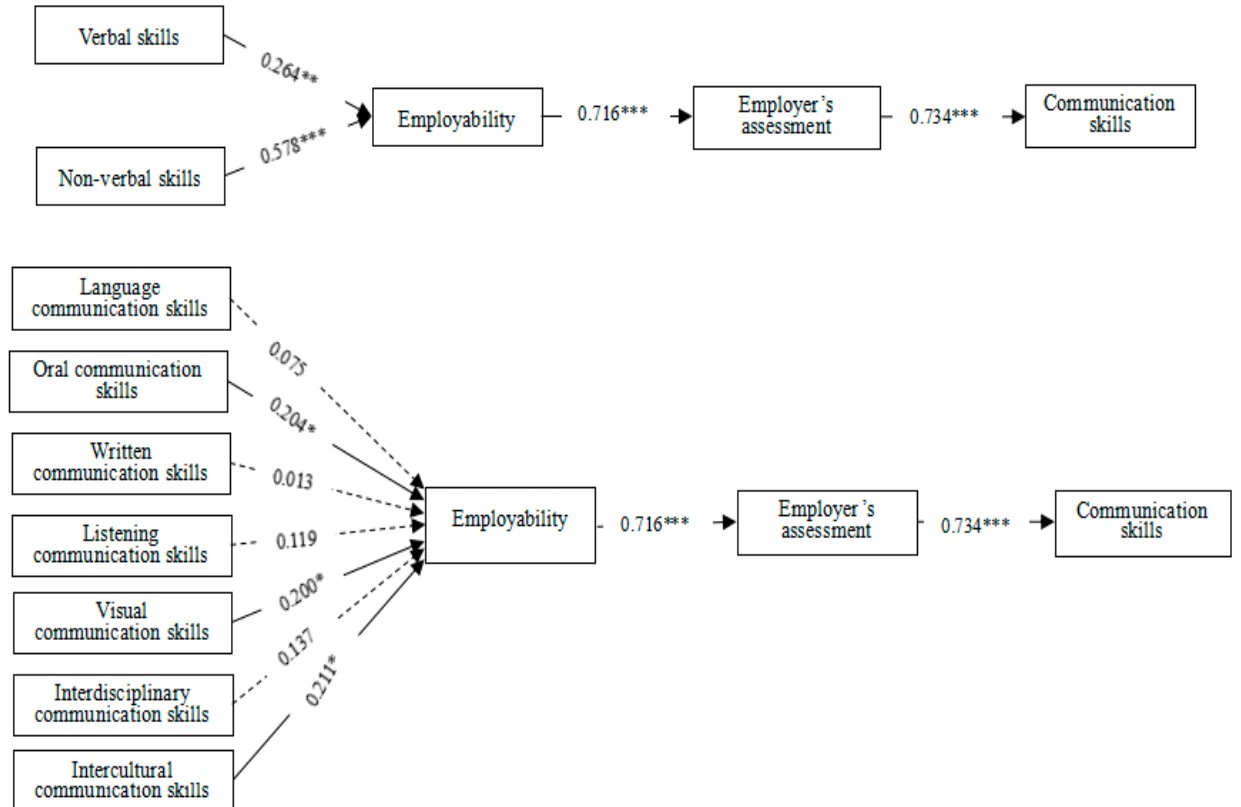

**Figure 4.** Path analysis for SEM. * $p < 0.05$; ** $p < 0.01$; *** $p < 0.001$.

In the results shown in Figure 4, verbal skills (β = 0.264, $p < 0.01$) and non-verbal skills (β = 0.578, $p < 0.001$) are positively related to employability. Employability (β = 0.716, $p < 0.001$) is positively related to the employer's assessment. Employer's assessment (β = 0.734, $p < 0.001$) is positively related to communication skills. Oral communication

skills ($\beta$ = 0.204, *p* < 0.05), visual communication skills ($\beta$ = 0.200, *p* < 0.05), and intercultural communication skills ($\beta$ = 0.211, *p* < 0.05) are related to employability. Language communication skills, written communication skills, listening communication skills, and interdisciplinary communication skills are not related to employability. So, H1, H1b, H2, H2b, H2d, H3, and H4 are supported, and H1a, H1c, H2a, and H2c are rejected.

In summary, this study tested the role of communication skills on employability through use of a questionnaire survey of 130 engineering students and engineering practitioners, and used regression analysis and SEM to test research hypotheses H1–H4. The results of the hypothesis testing are summarized in Table 4.

**Table 4.** Summary of hypothesis testing results.

| Hypothesis | Content | Result |
|---|---|---|
| H1 | Verbal skills have a positive effect on the employability of engineering students | Supported |
| H1a | Language communication skills have a positive effect on the employability of engineering students | Not Supported |
| H1b | Oral communication skills have a positive effect on the employability of engineering students | Supported |
| H1c | Written communication skills have a positive effect on the employability of engineering students | Not Supported |
| H2 | Non-verbal skills have a positive effect on the employability of engineering students | Supported |
| H2a | Listening communication skills have a positive effect on the employability of engineering students | Not Supported |
| H2b | Visual communication skills have a positive effect on the employability of engineering students | Supported |
| H2c | Interdisciplinary communication skills have a positive effect on the employability of engineering students | Not Supported |
| H2d | Intercultural communication skills have a positive effect on the employability of engineering students | Supported |
| H3 | Engineering students' employability has a positive effect on employer's assessment | Supported |
| H4 | Employer's assessment positively moderates the role of engineering students' communication skills on employability. | Supported |

## 6. Discussion

The findings of the empirical study highlight the positive role and importance of communication skills in the employability of engineering students. From the data analysis, it can be concluded that communication skills have a positive effect on engineering students' employability, while their employability influences the employer's assessment, which in turn positively influences students' communication skills, thereby effectively forming a closed-loop process.

(1) This study found that verbal skills (*p* < 0.01) and non-verbal skills (*p* < 0.001) based on the OBE theory directly and positively affect the employability of engineering students. This result supports the hypothetical model of communication skills–outcome-based education (CS-OBE) and enhances the explanatory validity of the model. It is also in line with the findings of Rasul et al. and Ihmeideh et al., which highlight the relationship between employability and good communication skills of engineering students, meaning good communication skills can help to engineering students to be more competitive in employment prospects [20,75]. They found that a positive communication environment provided students with opportunities to learn how to communicate and thus have better communication skills in order to be better prepared for the job market at the end of their studies.

(2) Verbal communication skills of communication through oral communication skills (*p* < 0.05) influenced the employability of engineering students. This result is in line with Mahbub et al., indicating that students' verbal communication skills are positively related to their employability in the learning and workplace [76]. High levels of oral communication skills have a positive impact on the career development of engineering students,

facilitating their professional knowledge and enhancing their teamwork skills to further solve professional problems. In addition, non-verbal communication skills influenced the employability of engineering students through visual communication skills ($p < 0.05$) and intercultural communication skills ($p < 0.05$). The results are also in agreement with the research findings of Feise and Schaer that the digital age places new demands on the communication skills of engineering students [51]. For example, good graphic and intercultural communication skills of engineering students in the digital age facilitate effective communication with stakeholders, improving students' ability to learn about other cultures, correspondingly increasing engineering students' learning and career opportunities, as well as broadening their career prospects.

(3) The study also identified that students' employability ($p < 0.001$) influenced the employer's assessments, which confirmed the previous studies of Gasova et al. [77]. As the employment requirements of students in the digital age continue to rise, employers' assessments are not limited to student technical skills but are demanding more from students' communication skills. The higher the ability to apply the dimensions in communication skills, the more competitive the student's employment prospects will be and the higher the employer's assessment will be. In the meantime, an improvement in the students' communication skills will be achieved. This is the key to promoting lifelong learning in their studies and work.

(4) At the same time, the employer's assessment ($p < 0.001$) simultaneously made a significant contribution to the further improvement of students' communication skills. This finding is consistent with the study by Mahbub et al. [76]. Employers recommend changes to university teaching methods, particularly in terms of exposing students to real or open-ended problems, to help them learn real problem-solving skills in a range of situations. Overton et al. designed an intervention for students that focuses on the development of skills needed for post-college-graduation success [78]. The program aims to involve employers in the development and implementation of such an intervention, which can gather evidence of effective teaching methods and activities through observation, interviews, and discourse analysis, ultimately enhancing students' communication skills to address the dilemmas they face in the workplace. Today's job market has become very competitive and, to be successful, candidates must have qualities that differentiate them from other candidates with similar qualifications. Engineering graduates need communication skills to remain relevant in a global environment.

(5) In addition to the hypotheses analyzed above based on OBE theory, the study found that language communication skills and written communication skills in verbal skills and listening communication skills, and interdisciplinary communication in non-verbal skills, did not have a significant effect on employability. This may be since 73.08% of the samples were enrolled engineering students and only 26.92% were already working, which indicates that the majority of the samples had not been involved in the projects in the field and that the students were not fully aware of the practical aspects of their future work as most institutions did not train them in the skills required for the projects. This result is consistent with the findings of Picard et al. that students can develop more employability skills during the project, but the projects used in teaching are very different in specification and scale to those involved in fieldwork, resulting in a discrepancy between students' perceptions of employment and the actual job requirements [79]. These non-significant hypotheses also have some practical implications, as institutions could use some projects to develop students' employability skills and equip them with the skills needed for employment such as capstone projects, thus promoting their communication and interdisciplinary skills.

The results of the study indicate the importance of communication skills for engineering students. Therefore, the results have practical implications for the development of engineering students, including the following recommendations:

(1) Offer courses and programs to enhance the language skills of engineering students. Educators should use market demand as a starting point for changes to existing curricula to improve students' skills. For example, English for Specific Purposes (ESP)

and writing courses could be established to focus engineering students on the specific terminology and communication skills required in their field of study and to improve their oral and written skills, thus contributing to their overall competence.

(2) Promote the use of digital media and offer digital courses. The enhancement of non-verbal skills is closely linked to digitalization, and relevant educators should transform traditional teaching methods, relying on digital media to bring students up to date, so as to continually enhance digital training for engineering students, strengthen the learning ability of engineering students in the use of technologies such as multimedia and virtual laboratories, and provide extensive courses on big data analysis to enhance students' communication skills in the digital age.

(3) Strengthen the practical application ability of graduating students. Relevant educational institutions and educators should collaborate with relevant enterprises to realize the integration of industry and education so that the education and training of colleges and universities can be precisely aligned with the practical ability of students, which allows students to improve their communication skills from actual project participation, and the trial-and-error cost of future employment of engineering students by "attracting enterprises to schools and students to enterprises" can be reduced.

## 7. Conclusions, Limitations, and Future Work

This study evaluated the role of engineering students' communication skills on their employability in China. The study is based on OBE theory and classified communication skills into verbal and non-verbal communication skills by reviewing the literature, and constructed a CS-OBE hypothesis model based on the key elements of OBE theory. According to on this study's findings, the CS-OBE model was investigated and tested using questionnaires, regression analysis, and SEM. The results show that:

(1) Communication skills, as one of the important soft skills, affect the employability of engineering students through verbal skills and non-verbal skills. When engineering students have good oral and visual communication skills required by the digital age, and the ability to avoid cultural conflicts in transnational cooperation, they can gain more discourse rights and opportunities in the workplace, which can significantly enhance their employment competitiveness.

(2) The employability of engineering students also affects the employer's assessment. The change in digital times puts forward higher requirements for the employability of engineering students. Therefore, the cultivation of modern students should not only be limited to the improvement of professional skills but should be closer to the cultivation of compound talents. This training mode will further change the employer's assessment from a single technical-level assessment to a comprehensive quality assessment that conforms to the overall development of the job market.

(3) Employer's assessment further promotes the improvement of communication skills of engineering students. Employers believe that, with the continuous development of science and technology, engineering students should understand the use of various media, have the communication skills necessary for the workplace, such as listening, speaking, reading, and writing, and require engineering education institutions to adopt corresponding training strategies to develop workers who meet the job market.

The study had certain limitations. Firstly, due to the time and space constraints of the study, the scope of the survey was limited to engineering students in higher education in China, and there was a lack of participation from university teachers and a relatively small number of questionnaires from practitioners. Secondly, most of the participants were enrolled engineering students and the sample size of working graduates was relatively small. Therefore, in the future, consideration should be given to obtaining research data from teachers and graduates from different regions and schools to further corroborate the findings using a larger sample size and a wider range of participants.

**Author Contributions:** Writing—original draft, Y.W. and L.X.; Writing—review & editing, S.P.P. All authors have read and agreed to the published version of the manuscript.

**Funding:** This research was funded by [Integrated Built Environment Teaching & Learning in the Joint Curriculum Development amid Digital-Driven Industry 4.0 among China, Vietnam, and UK].

**Institutional Review Board Statement:** Not applicable.

**Informed Consent Statement:** Not applicable.

**Data Availability Statement:** Not applicable.

**Conflicts of Interest:** The authors declare no conflict of interest.

## Appendix A

**Table A1.** The scale of testing the role of engineering students' communication skills on employability.

| Variable | | Item Labels | Items | Source |
|---|---|---|---|---|
| Verbal Skills | Language communication skills | LCS1 | With increasing internalization and digital communication platforms widely used, it is very important to master a foreign language skill. | [80,81] (Kovac and Sirkovic 2017; Byrne, Weston, and Cave 2020) |
| | | LCS2 | The inability to use English skills proficiently in study or work can affect communications and interactions with international engineering students. | |
| | | LCS3 | Poor English skills have led to a fear of attempting to participate in international engineering teamwork projects. | |
| | | LCS4 | The inability to use English proficiently has resulted in the weakness to read professionally relevant foreign language literature smoothly and cannot enhance my knowledge. | |
| | Oral communication skills | OCS1 | It is important to communicate and speak regularly with others in my work or study. | [3,82] (Ajit and Deshmukh 2013; Bee and Hie 2015) |
| | | OCS2 | Inadequate oral communication skills can lead to an inability to express my ideas clearly. | |
| | | OCS3 | Inability to communicate smoothly with others can result in poor learning or productivity. | |
| | | OCS4 | PPT presentations, group discussions, class discussions and role-playing will enhance my oral communication skills. | |
| | Written communication skills | WCS1 | Writing communication skills are important in study or work. | [3] (Bee and Hie 2015) |
| | | WCS2 | It is important to use digital media (WeChat, email, etc.) to express my views and ideas accurately. | |
| | | WCS3 | Engineering students are required to have good writing skills (including writing lab reports, technical writing and essays, etc.). | |

**Table A1.** *Cont.*

| Variable | Item Labels | Items | Source |
|---|---|---|---|
| | LSCS1 | When using digital communication software and digital learning platforms, having good listening skills can enhance learning or productivity. | [82,83] (Ajit and Deshmukh 2013; Kenayathulla, Ahmad, and Idris 2019) |
| Listening communication skills | LSCS2 | Active listening helps to get to the speaker's ideas when listening to others and to summarize what is important and to apply it. | |
| | VCS1 | Visual expressions such as images, icons and images can simplify and improve communication. | [28] (Riemer 2007) |
| Visual communication skills | VCS2 | Digital engineering drafting software helps to communicate effectively and efficiently with professionals. | |
| | IDCS1 | The deepening of the digitalization process requires the integration and convergence of relevant technologies, and a lack of knowledge of other expertise can affect the effectiveness of disciplinary cooperation and lead to communication barriers. | [84] (Hart 2019) |
| Interdisciplinary communication skills | IDCS2 | Interdisciplinary communication enhances teamwork skills in study or work. | |
| | IDCS3 | Interdisciplinary communication can help to become a versatile talent, thus increasing learning and employment opportunities. | |
| | ICCS1 | How to use digital communication and learning platforms to collaborate across countries and cultures is crucial to the career development of an engineer. | [81] (Byrne, Weston, and Cave 2020) |
| Intercultural communication skills | ICCS2 | Learning about the culture of another country and understanding the development of the engineering industry in another country is extremely useful in enhancing professional competence. | |
| | EM1 | Having good communication skills can contribute to employability. | [80] (Kovac and Sirkovic 2017) |
| Employability | EM2 | The impact of the digital age is such that a lack of communication skills can lead to the loss of many job opportunities. | |
| | EM3 | To be a successful engineer, I need to have good communication skills. | |
| | EA1 | The rapid development of the digital age has led to an increasing demand for communication skills from employers. | [52,85] (Husain et al. 2010; Okolocha and Odimmega 2019) |
| Employer's assessment | EA2 | Employer's assessment can promote further learning and thus enhance the employability of students. | |

The "Non-verbal Skills" label spans the Listening, Visual, Interdisciplinary, and Intercultural communication skills rows in the Variable column.

## Appendix B

**Table A2.** Descriptive statistics and correlation analysis. The descriptive statistics and correlation of studied variables.

| | Mean | SD | 1 | 2 | 3 | 4 | 5 | 6 | 7 | 8 | 9 |
|---|---|---|---|---|---|---|---|---|---|---|---|
| 1. Language communication skills | 4.337 | 0.717 | (0.911) | | | | | | | | |
| 2. Oral communication skills | 4.283 | 0.565 | 0.647 *** | (0.825) | | | | | | | |
| 3. Written communication skills | 4.308 | 0.588 | 0.698 *** | 0.749 *** | (0.851) | | | | | | |
| 4. Listening communication skills | 4.412 | 0.576 | 0.662 *** | 0.742 *** | 0.726 *** | (0.912) | | | | | |
| 5. Visual communication skills | 4.308 | 0.569 | 0.557 *** | 0.650 *** | 0.681 *** | 0.688 *** | (0.937) | | | | |
| 6. Interdisciplinary communication skills | 4.356 | 0.607 | 0.640 *** | 0.643 *** | 0.776 *** | 0.756 *** | 0.720 *** | (0.899) | | | |
| 7. Intercultural communication skills | 4.338 | 0.697 | 0.643 *** | 0.622 *** | 0.791 *** | 0.665 *** | 0.606 *** | 0.815 *** | (0.968) | | |
| 8. Employability | 4.385 | 0.551 | 0.629 *** | 0.700 *** | 0.714 *** | 0.711 *** | 0.691 *** | 0.732 *** | 0.708 *** | (0.877) | |
| 9. Employer's assessment | 4.442 | 0.643 | 0.587 *** | 0.560 *** | 0.683 *** | 0.594 *** | 0.579 *** | 0.685 *** | 0.715 *** | 0.716 *** | (0.959) |

Note: The coefficients represent in the diagonal are the square root value of AVE. SD = standard deviation. *** $p < 0.001$.

## Appendix C

**Table A3.** Results of CITC and alpha-coefficient analysis of study variables.

| Dimension | Variable | Corrected Item and Total Correlation | Alpha |
|---|---|---|---|
| LSC | LCS1 | 0.692 | 0.930 |
| | LCS2 | 0.730 | |
| | LCS3 | 0.764 | |
| | LCS4 | 0.739 | |
| OCS | OCS1 | 0.683 | 0.833 |
| | OCS2 | 0.733 | |
| | OCS3 | 0.699 | |
| | OCS4 | 0.546 | |
| WCS | WCS1 | 0.739 | 0.808 |
| | WCS2 | 0.753 | |
| | WCS3 | 0.732 | |

**Table A3.** *Cont.*

| Dimension | Variable | Corrected Item and Total Correlation | Alpha |
|---|---|---|---|
| LSC | LSCS1 | 0.782 | 0.799 |
| | LSCS2 | 0.730 | |
| VSC | VCS1 | 0.744 | 0.861 |
| | VCS2 | 0.690 | |
| IDCS | IDCS1 | 0.760 | 0.881 |
| | IDCS2 | 0.759 | |
| | IDCS3 | 0.786 | |
| ICCS | ICCS1 | 0.784 | 0.931 |
| | ICCS2 | 0.827 | |
| EM | EM1 | 0.762 | 0.863 |
| | EM2 | 0.764 | |
| | EM3 | 0.697 | |
| EA | EA1 | 0.721 | 0.911 |
| | EA2 | 0.740 | |

**Appendix D**

**Table A4.** KMO and Bartlett's Test.

| KMO and Bartlett's Test | | |
|---|---|---|
| KMO | | 0.922 |
| Bartlett's test of sphericity | Approx. Chi-square | 3146.108 |
| | df | 300 |
| | Sig. | 0.000 |

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
