# Peer review of "Evaluating the Role of the Communication Skills of Engineering Students on Employability According to the Outcome-Based Education (OBE) Theory"

_sustainability, doi:10.3390/su15129711_

Round 1

Reviewer 1 Report

The Manuscript titled “Evaluating the Role of the Communication Skills of Engineering Students on Employability According to the OBE Theory”, generally presented good quality research material in highlighting, the hypothesis, and methodological statement. The selected problem of the topic and the point of the view on analyzing the problem is interesting. Several comments regarding the improvement of the manuscript are 

following: 

1. The part of literature review, it is suggestive to add more latest references published in 2023 relevant to the research.

2. In data collection, please provide additional details to clarify if the informed consent has been specified in online submission information. 

3. The discussion section lack comparison with previous literature.

4. The references are okay; however, there are still a few inappropriate references. Kindly update them. 

5. The Conclusion section needs to include theoretical and practical contributions. And the practical suggestions that you make need to be concrete.

No.

Author Response

Dear Respected Editor and Reviewers,

Thank you very much for the valuable comments and suggestions on our manuscript entitled “Evaluating the Role of the Communication Skills of Engineering Students on Employability According to OBE Theory”. According to the comments, we have revised our manuscript carefully and red-marked all the revised parts in the edited version. The suggestions have helped us improve the quality of our manuscript to a higher level. Below is our point-to-point response to the Editor/Reviewers comments. In addition, we have checked the format to meet journal requirements. We deeply appreciate your consideration of our manuscript and look forward to your reply. Thanks very much!

Best Regards,

All Authors

Reviewer 2 Report

I’ve read with interest this manuscript that evaluates the influence of communication skills on the employability of engineering students from the aspects of verbal and non-verbal skills through the four key elements of OBE theory. The authors define and present the problem in an adequate way. The topic is important and advances in this field are very useful to the employability of engineering students nowadays. The experimental design is appropriate to test the hypothesis. Interview questions are listed in Appendix 1 and explained, their validity and reliability were tested. Tables and statistical indicators help the reader follow the discussion. There is no doubt that the investigation was well-conducted, however the manuscript is mostly centered in quantitative analysis. But I need a good description of the instruments and the procedures followed by the investigators during the collection of data; it would be helpful to see a presentation/discussion about a few examples, so the reader can also see some positive and/or negative aspects that were found in this investigation.

.       I would recommend to think on the title of the article. The usage of abbreviation OBE without any indication on it is unacceptable in the title.  I managed to decode it only when I’ve read line 65.

It is not clear for me whether the authors refer communicative skills to the professional hard skills of engineers or to the soft skills.  I also noticed that the communicative competence is viewed rather narrow/ What about the importance to use programming languages? The authors mention digital communication only lightly. In my opinion, this aspect should be incorporated in the understanding of communicative competence as well as the application of communicative skills in professional context.

The consideration by the authors of these few recommendations will make this manuscript a more solid piece for this academic journal.

Author Response

Dear Respected Editor and Reviewers,

Executive Editor,

Thank you very much for the valuable comments and suggestions on our manuscript entitled “Evaluating the Role of the Communication Skills of Engineering Students on Employability According to OBE Theory”. According to the comments, we have revised our manuscript carefully and red-marked all the revised parts in the edited version. The suggestions have helped us improve the quality of our manuscript to a higher level. Below is our point-to-point response to the Editor/Reviewers comments. In addition, we have checked the format to meet journal requirements. We deeply appreciate your consideration of our manuscript and look forward to your reply. Thanks very much!

Best Regards,

All Authors
